# Cell Type-Specific Anti-Viral Effects of Novel SARS-CoV-2 Main Protease Inhibitors

**DOI:** 10.3390/ijms24043972

**Published:** 2023-02-16

**Authors:** Nina Geiger, Viktoria Diesendorf, Valeria Roll, Eva-Maria König, Helena Obernolte, Katherina Sewald, Julian Breidenbach, Thanigaimalai Pillaiyar, Michael Gütschow, Christa E. Müller, Jochen Bodem

**Affiliations:** 1Institute for Virology and Immunobiology, University of Würzburg, Versbacher Strasse 7, 97078 Würzburg, Germany; 2Member of DZL, BREATH and iCAIR, 30625 Hannover, Germany; 3Pharmaceutical Institute, Pharmaceutical & Medicinal Chemistry University of Bonn, An der Immenburg 4, 53121 Bonn, Germany

**Keywords:** SARS-CoV-2, protease inhibitors, cell line specificity pyridyl indole carboxylates, azapeptide nitriles, peptidomimetics

## Abstract

Recently, we have described novel pyridyl indole esters and peptidomimetics as potent inhibitors of the severe acute respiratory syndrome coronavirus type 2 (SARS-CoV-2) main protease. Here, we analysed the impact of these compounds on viral replication. It has been shown that some antivirals against SARS-CoV-2 act in a cell line-specific way. Thus, the compounds were tested in Vero, Huh-7, and Calu-3 cells. We showed that the protease inhibitors at 30 µM suppress viral replication by up to 5 orders of magnitude in Huh-7 cells, while in Calu-3 cells, suppression by 2 orders of magnitude was achieved. Three pyridin-3-yl indole-carboxylates inhibited viral replication in all cell lines, indicating that they might repress viral replication in human tissue as well. Thus, we investigated three compounds in human precision-cut lung slices and observed donor-dependent antiviral activity in this patient-near system. Our results provide evidence that even direct-acting antivirals may act in a cell line-specific manner.

## 1. Introduction

Since the end of 2019, SARS-CoV-2, the virus that causes the coronavirus disease of 2019 (COVID-19), has become a pandemic threat, with more than 1.7 million deaths after one year and more than 6.7 million deaths by January 2023 [1]. Despite the fast development of vaccines, which can prevent severe outcomes even in high-risk patients, the number of lethal infections still increases. While all countries implemented vaccination programs, only some reached levels for the first vaccination above 90%. Seven countries including Yemen, Haiti, and Cameroon, could still not vaccinate more than 10% of their population. Seventy-four countries have failed to vaccinate 50% of their population, leaving the need to develop effective antiviral therapies [1].

In the past, the development of antivirals has led to drugs that prolonged the lifespan of infected patients, e.g., in the case of human immunodeficiency virus-1 (HIV-1) by decades, or even cured infected patients entirely within weeks, as in hepatitis C virus (HCV) infections. Direct-acting antivirals inhibit viral entry or essential viral enzymes such as proteases, polymerases, integrases, terminases, and helicases (reviewed in [2,3]). In both HCV and HIV-1, mono- or dual-therapies resulted in the generation of resistant viral strains. For HIV-1, the selection of resistant viruses was due to incomplete suppression of viral replication [4]. This indicates that viruses replicating with high viral loads require antiviral combination therapies, suppressing viral loads by 6 or more orders of magnitude [4]. Unfortunately, SARS-CoV-2 belongs to the group of viruses where more than 10^6^ genome copies can be found in samples from infected patients.

Recently, we and others have reported the development of compounds inhibiting the SARS-CoV-2 main protease (M^pro^) with IC_50_ values in the low nanomolar to micromolar range [5,6]. These protease inhibitors (Figure 1) bear an electrophilic warhead, susceptible to the nucleophilic attack of the active-site cysteine residue of the main protease, which leads to covalent enzyme-inhibitor complexes. Compounds **1**–**10** are pyridyl esters whose mode of action involves the acylation of the active site cysteine residue, whereupon a pyri-din-3-olate acts as a leaving group. The resulting acyl-enzyme undergoes slow hydrolysis, which may lead to a pseudo-irreversible inhibition of the protease [5,7,8]. Crystallographic evidence for such a covalent mechanism has been obtained for structurally related M^pro^ inhibitors [9]. Compounds **11**–**20** were designed based on the peptidic structure of substrates of the SARS-CoV-2 main protease [10] and equipped with a nitrile warhead as part of an aza-amino acid. The strong and irreversible inhibition of human [11], schistosomal [12], and viral cysteine proteases [5] by azapeptide nitriles is based on the formation of covalent isothiosemicarbazide adducts.

Here, we analyse whether our protease inhibitors block SARS-CoV-2 replication in cell culture. Since recent experiments have indicated that the sensitivity of SARS-CoV-2 to antiviral compounds depends on the employed cell line [13], we sought to analyse the compounds’ efficiencies on the three most common cell lines, namely Vero cells, Huh-7 cells, and Calu-3 cells infected with human SARS-CoV-2, and compare the results with precision-cut lung slices (PCLS), a patient-near assay system for SARS-CoV-2 testing.

## 2. Results and Discussion

Because antiviral compound screens against SARS-CoV-2 are sensitive to the cell lines used for the assays, we initially decided to determine the influence of our protease inhibitor library against SARS-CoV-2 in Vero and Huh-7 cells first. In a second step, all positive compounds were re-analysed in Calu-3 cells. The inhibition of SARS-CoV-2 was assessed by incubating Vero cells with the protease inhibitors at decreasing concentrations.

Before analysing the effects of the protease inhibitors on the SARS-CoV-2 replication, the potential cell toxicity of the compounds was investigated. Vero and Huh-7 cells were incubated with the compounds, and the cell growth rate was compared to untreated cells. In addition, we performed viability assays using 3-(4,5-dimethylthiazol-2-yl)-2,5-diphenyl-tetrazolium bromide (MTT) on these cell lines and Calu-3 cells, as described previously [14,15,16,17]. Concentrations reducing cell growth or catabolism by 15% or more were excluded from further analyses (Table 1). The highest sensitivity was observed with Calu-3 and Huh-7 cells, where six more compounds could not be used at concentrations of 10 µM or more, indicating that Vero cells tolerate higher concentrations (Table 1). The pyridin-3-yl indole-carboxylates **1** and **2** were toxic to Calu-3 cells but not to the other two cell lines.

Next, the inhibition of viral replication by the protease inhibitors was analysed. The cells were incubated with the compounds and subsequently infected with SARS-CoV-2 at a multiplicity of infection (MOI) of 1 and incubated for three days, as described before [14,15,17]. Viral stocks contain non-infectious viruses that remain in the supernatant and influence the quantification of the viral genome. Thus, the medium was replaced 24 h after infection by a medium containing the respective test compounds to remove these defective viruses, which would influence the determination of viral genome copies. The cell culture supernatants were collected 72 h after infection, the viral RNAs were isolated, and genome copy numbers were determined by quantitative reverse transcriptase polymerase chain reaction (RT-qPCR). As expected, the observed suppression of viral replication did not correlate with the IC_50_ values determined in the in vitro enzyme assays (Table 1 and data in [5] since uptake and export limit intracellular drug concentrations.

The protease inhibitors **4** and **5**, which contain a fused thiophene or furan ring (Figure 1), failed to suppress SARS-CoV-2 replication in all three cell lines (Table 1). The analogous indole derivative **1** was only moderately active. In contrast, the chloro-substituted inhibitors **2** and **3** showed better antiviral activity, indicating the advantageous electron-withdrawing effect of the substituents, which increases the electrophilicity at the ester carbonyl carbon (Figure 1 and Table 1). Disulfiram, used as a control, was effective in Vero and Huh-7 cells, similar to the recently published data with 293T-ACE2 and Vero E3 cells [18].

Of note, among the pyridin-3-yl indole-carboxylates, we identified three protease inhibitors to be active as antiviral compounds in the three different cell lines. They inhibited viral replication in Vero, Huh-7, and Calu-3 cells at a concentration of 30 µM by up to 5.3 orders of magnitude (Table 1). The common feature of the compounds **6**, **7**, and **9** is the presence of an unsubstituted indole core with the halopyridinyl carboxylate moiety at the 4- or 7-positon of the indole. The structurally related halopyridinyl indole esters **3** and **8**, connected in the 2- or 5-position of the indole ring system, prevented viral replication in Huh-7 and Calu-3 cells but could not reduce genome copies by more than one order of magnitude in Vero cells. These results underline the importance of characterising antiviral compounds in more than one cell line. The tricyclic tetrahydro-1*H*-carbazole ester **10** did not reduce viral loads in any of the applied cell lines.

The cell line dependency of viral inhibition is surprising but might be explained by different bioavailability in the different cell lines. Since certain indole esters were found to inhibit SARS-CoV-2 in several cell lines, those inhibitors might serve as new lead structures for further development, including animal experiments.

Next, the protease inhibitors with a peptide-like structure were analysed. Again, the first step was to determine the cytotoxicity of the candidates in the different cell lines. The peptidomimetic compounds **13** and **14** exhibited a cytotoxic effect on two cell lines tested and were excluded from further analyses. The compound **12** inhibited the viral replication in Vero and Calu-3 cells but was cytotoxic on Huh-7 cells. The azapeptide nitriles **11**, **16**, and **18** suppressed viral replication in Huh-7 cells. Inhibitors **15**, **17**, **19**, and **20** did not reduce viral replication in all three cell lines.

Since the dipeptide derivative **16** reduced viral titers in Vero cells by more than four orders of magnitude, it was selected for further investigation, together with the potent indole esters **6** and **7**, which suppressed viral titers by more than 2 log units in Vero and Huh-7 cells. We evaluated these three inhibitors at different concentrations and determined EC_50_ values of 5.8 µM for **6**, 9.95 µM for **7**, and 2.4 µM for **16** in Vero cells (Table 1). The value for **6** was consistent with a published EC_50_ value for this compound of 2.8 µM, previously determined in VeroE6 cells [19].

Next, we sought to confirm the results of selected active protease inhibitors, **7**, **8**, and **16**, in human precision-cut lung slices (PCLS) since they represent a patient-near infection system. PCLS from two individual donors were incubated with the compounds and subsequently infected with SARS-CoV-2 (MOI 10) for 72 h. The viral infectivity was quantified by infecting Vero cells with the cell culture supernatants for 72 h. The viral replication was determined by RT-qPCR (Figure 2). In donor 1, compounds **7** (a pyridyl ester) and **16** (a peptidomimetic) suppressed viral replication by approximately 4.5 orders of magnitude. However, the inhibition by the compounds was donor-dependent, and one of the inhibitors, compound **16**, which could be tested in a second patient sample, was much less active in the case of donor 2. Pyridyl ester **8**, which could only be tested in patient 2, was somewhat more active than **16**. These results prove that the analysed inhibitors could suppress viral replication in patients, but the effect may be patient-dependent. The selection of inhibitors active in more than one cell line led to active compounds in human tissue.

In summary, we have shown that the characterisation of antiviral drugs directed against SARS-CoV-2 relies on the cell culture system. In Calu-3 cells, viral replication was less susceptible to inhibition by several compounds compared to the effects in Vero and Huh-7 cells. This could be the reason for a better predictive value of the Calu-3 cells regarding active drugs in vivo. If Calu-3 cells exhibit a lower drug uptake, which resembles the in vivo tissue, the selection of compounds in these cells more likely leads to active compounds in patients. However, this hypothesis needs to be evaluated in the future.

## 3. Materials and Methods

### 3.1. Viral Infection and RNA Quantification

The virus isolate has been described before [14,20]. The cells were seeded in 48-well plates (Vero, 15,000/well; Huh-7, 30,000/well; Calu-3, 100,000/well). The next day the cells were incubated with the compounds and, after 10 to 15 min, infected with SARS-CoV-2. After 24 h, the medium was exchanged to remove inactive viruses since they would influence the genome copy determination. All infection experiments were performed in triplicate assays and repeated at least twice in independent experiments. After 72 h, 200 µL of the medium was collected, and viral genomes were purified with the High Pure Viral Nucleic Acid kit (Roche, Mannheim, Germany). SARS-CoV-2 RNA genomes were quantified with the dual-target SARS-CoV-2 RdRP RTqPCR assay kit, containing universal SARS-CoV-2 primers and with viral RNA multiplex master kit (Roche, Mannheim, Germany) with the LightCycler 480 II (Roche, Mannheim, Germany). The provided standard was used for genome copy-number quantification using the LightCycler 480 II Software Version 1.5 (Roche, Mannheim, Germany). The PCR reactions were performed in triplicates. Quantifications were performed with the respective cycler software. EC_50_ values were calculated using GraphPad Prism Version 6 (GraphPad Software; Boston, MA, USA).

### 3.2. Cytotoxicity and Cellular Proliferation Assays

The proliferation of cells was determined by direct automatic cell counting. Cells were seeded on optical plates (Vero, 3500/well; Huh-7, 5000/well) (CellCarier-96, Perkin-Elmer, Waltham, MA, USA) and counted before the experiments. The compounds were added at decreasing concentrations from 60 µM, and the cells were incubated for 72 h, similar to the infection time. The cell numbers per well were determined using the PerkinElmer Ensight reader. The numbers were compared to those of the solvent controls. All experiments were performed in six independent assays in parallel, and the standard deviation was calculated.

Cytotoxicity of the compounds in Calu-3 cells was determined by MMT assays (Promega, Mannheim, Germany) since Calu-3 cells cannot be reliably counted with the PerkinElmer Ensight reader. Calu-3 cells (20,000/well) were seeded in 96-well plates. The next day, the compounds were added, and the cells were incubated for 72 h. Then, 10 µL of the MTT substrate was added, and the absorbance was measured after 1 h of incubation at 37 °C.

### 3.3. Human Precision-Cut Lung Slices

Infection of human precision-cut lung slices (PCLS) and determination of viral infections were performed as described before [14]. After the transport, human PCLS were incubated for 1 h at 37 °C in DMEM/F12 medium (Life Technologies, Darmstadt, Germany) supplemented with 1% Penicillin/Streptomycin (Lonza, Verviers, Belgium) and separated on a 48-well dish. The compounds were added, and the PCLS were infected with SARS-CoV-2 with a high MOI of approximately 10 for 3 days. Viral infectivity was determined by infecting Vero cells with 100 µL of the cell culture supernatants in duplicates. Viral genomes were determined after 72 h of infection by RTqPCR.

## Figures and Tables

**Figure 1 ijms-24-03972-f001:**
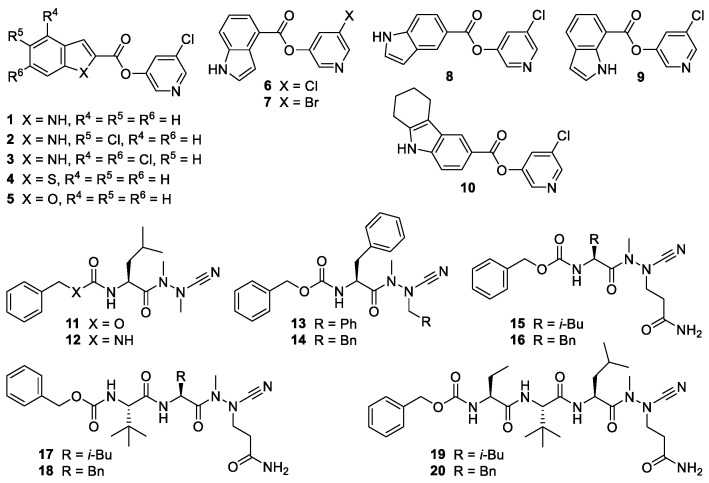
Chemical structures of the investigated protease inhibitors.

**Figure 2 ijms-24-03972-f002:**
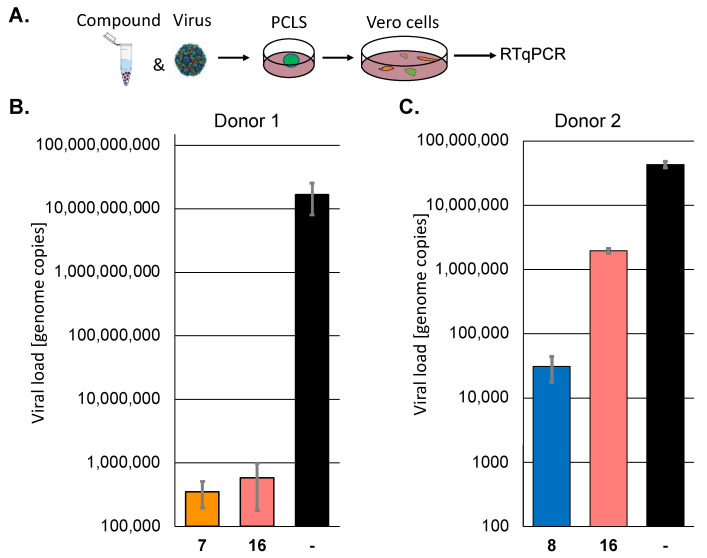
The protease inhibitors block SARS-CoV-2 replication in human PCLS of two different donors. (**A**) Scheme of the experimental setup. (**B**,**C**) Human PCLS were incubated with 30 µM of selected protease inhibitors and infected with SARS-CoV-2. All infections were performed in triplicates. Viral infectivity was determined by infecting Vero cells in duplicates with the supernatants. Viral loads were determined by RT-qPCR.

**Table 1 ijms-24-03972-t001:** Inhibition of SARS-CoV-2 dependent on the infected cell line.

Compound Class	Compound	Antiviral Activity ^a,b^			EC_50_ ^c^ [µM]
		Vero	Huh-7	Calu-3	
Disulfiram		1.5 ± 0.7	2.0 ± 0.4	0.2 ± 0.1	
Pyridin-3-yl 1*H*-indole-2-carboxylates and analogs	**1**	0.6 ± 0.2	0.8 ± 0.2	n.a.	
	**2**	1.6 ± 0.4	2.8 ± 1.3	tox	
	**3**	0.2 ± 0.2	2.0 ± 0.2	1.1 ± 0.4	
	**4** ^d^	0.1 ± 0.1	0.5 ± 0.3	n.a.	
	**5** ^d^	0.1 ± 0.1	0.0	0.0	
Pyridin-3-yl-1*H*-indole-4-, 5-, or 7-carboxylates and analogs	**6**	3.2 ± 0.6	4.2 ± 0.0	1.0 ± 0.7	5.8
	**7**	2.8 ± 1.0	5.3 ± 0.1	1.2 ± 0.3	9.9
	**8**	0.6 ± 0.3	2.4 ± 0.1	2.0 ± 0.6	
	**9**	3.8 ± 0.3	3.6 ± 0.5	1.7 ± 0.7	
	**10** ^e^	0.0	0.3	n.a.	
Azapeptide nitriles	**11**	3.3 ± 1.4	3.2 ± 0.8	0.0	
	**12**	4.6 ± 0.2	tox	2.4 ± 1.0	
	**13**	n.a.	tox	tox	
	**14**	n.a.	tox	tox	
	**15**	0.0	0.5 ± 0.1	0.0	
	**16**	4.1 ± 0.7	1.0 ± 0.1	n.a.	2.4
	**17**	0.0	0.0	0.0	
	**18**	0.0	1.6 ± 0.6	0.1	
	**19**	0.0	0.0	0.3	
	**20**	0.0	0.8	0.1	

^a^ Reduction in logarithmic scale compared to the control. Depicted is the mean from at least three independent experiments performed in triplicate assays. Tox: substance showed toxicity; n.a.: not analysed. ^b^ Compounds were tested at a concentration of 30 µM. ^c^ The EC_50_ was determined on Vero cells. ^d^ Benzo[*b*]thiophene and benzofuran derivatives, respectively. ^e^ 2,3,4,9-Tetrahydro-1*H*-carbazole derivative.

## Data Availability

Not applicable.

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
