# Peer review of "Cell Type-Specific Anti-Viral Effects of Novel SARS-CoV-2 Main Protease Inhibitors"

_ijms, 2023, doi:10.3390/ijms24043972_

Round 1

Reviewer 1 Report

The paper entitled "Cell type-specific anti-viral effects of novel SARS-CoV-2 main protease inhibitors" describes the biological evaluation as antiviral of a panel of 20 SARS-CoV-2 Main Protease inhibitor against replication of SARS-CoV-2 inside several cell lines. In details, the authors selected the inhibitors from a previously published work by them and evaluated against SARS-CoV-2 in three different cell types, Vero, Huh-7 and Calu-3. The communication may represent a continuation of the previous work, which involved the discovery of the Main Protease inhibitors, belonging to pyridyl esters and azanitriles. The article is well written, easy to read and the results are shown in a proper manner, with the help of a clear table. I suggest some minor revision before publication in International Journal of Molecular Sciences.

-       1. The introduction is too concise. I suggest to expand the part about the Main Protease of SARS-CoV-2, adding some references and describing more in details the relevance of Mpro inhibitors in the discovery of new anticoronavirus drugs (Drug Discovery Today, 2021, 26.3: 804-816, Biomolecules, 2021, 11.4: 607).

-        2. It is not clear for me why the authors did not select pulmonary cell lines (like MCR-5) for the assay, considering that they are the ideal target for SARS-CoV-2. I suggest to have a look at this: Sci Rep. 2021 Mar 8;11(1):5376.

-        3. I suggest to add the EC50 values of compounds 6,7 and 16 in the Table 1.

Author Response

The introduction is too concise. I suggest to expand the part about the Main Protease of SARS-CoV-2, adding some references and describing more in details the relevance of Mpro inhibitors in the discovery of new anticoronavirus drugs (Drug Discovery Today, 2021, 26.3: 804-816, Biomolecules, 2021, 11.4: 607).

We added a new paragraph and references to the introduction, including the reference the reviewer suggested. However, we have no access to the paper in Drug Discovery Today. Thus, we could not include this reference.

-        2. It is not clear for me why the authors did not select pulmonary cell lines (like MCR-5) for the assay, considering that they are the ideal target for SARS-CoV-2. I suggest to have a look at this: Sci Rep. 2021 Mar 8;11(1):5376.

MCR-5 cells are lung fibroblasts and do not express the ACE2 receptor, while Calu-3 cells are epithelial lung cells commonly used with SARS-CoV-2. Calu-3 cells closely represent the human lung tissue and show, at least with Chloroquine and Fluoxetine, similar SARS-CoV-2 replication kinetics to PCLS.  

-        3. I suggest to add the EC50 values of compounds 6,7 and 16 in the Table 1.

We added this information to Table 1.

Reviewer 2 Report

I.        Abstract: “the protease inhibitors suppress viral replication by up to 5 orders of magnitude in Huh-7 cells, while in Calu-3 cells, suppression by 2 orders”. The concentration at which protease inhibitors displayed these effects should be mentioned.

II.        “Since recent experiments have indicated that the sensitivity of SARS-CoV-2 to antiviral compounds depends on the employed cell line”. Reference needed.

III.        In Table 1 footnote: “Tox: substance showed toxicity” and in the following sentence: “Concentrations affecting cell growth or catabolism were excluded from further analyses (Table 1). The highest sensitivity was observed with Calu-3 and Huh-7 cells, where six more compounds could not be used at concentrations of 10 µM or more, indicating that Vero cells tolerate higher concentrations (Table 1)”. Admittedly, it is reliable to exclude the cytotoxic effects of the studied compounds during the evaluation of their viral replication inhibitory activity. However, the authors have to clarify the cytotoxicity threshold (growth inhibition% per µM concentration of the tested compound) above which they considered “substance showed toxicity”.

IV.        Materials and Methods: “The cells were incubated with the compounds and subsequently infected with SARS-CoV-2. After 72 h ………” The time between incubating the compounds and infection with SARS-CoV-2 should be clarified. Was it the same as the time of incubation of the compounds for the preliminary cytotoxicity study??

V.        The viral replication was determined by RT-qPCR (Figure 2). There are two figures 1, please renumber the second figure.

VI.        The results (e.g. “The pyridin-3-yl indole-carboxylates 1 and 2 were toxic to Calu-3 cells but not to the other two cell lines. The protease inhibitors 4 and 5, which contain a fused thiophene or furan ring, failed to suppress SARS-CoV-2 …… Disulfiram, used as a control ……”) should be more connected, more determined and more thoroughly discussed. Referral to relevant figures or tables is preferred.

VII.        Summary: “In Calu-3 cells, viral replication was less susceptible to inhibition by several compounds ……. If Calu-3 cells exhibit a lower drug uptake, which resembles the in vivo tissue ….”. Is there any evidence or reference to be cited which supports the hypothesis of “Calu-3 cells exhibit a lower drug uptake”?

VIII.        Vero and Huh-7 cells were seeded in optical 96-well plates. The following day, the cell numbers per individual well 75 were determined with an Insight reader ……….. (MTT) on these cell lines and on Calu-3 cells as described previously. [6-9]”. Detailed experimental procedures should be transferred to the Materials and Methods section.

IX.        Re-editing for clarity enhancement may be beneficial, e.g. “The medium was replaced 24 h after infection by a medium containing the respective test compounds to remove defective viruses”.

X.        As expected, the observed suppression of viral replication did not correlate with the IC50 values determined in the in vitro enzyme assays (Table 1)”. Refer to the relevant Figure along with Table 1.

Author Response

  1. Abstract: “the protease inhibitors suppress viral replication by up to 5 orders of magnitude in Huh-7 cells, while in Calu-3 cells, suppression by 2 orders”. The concentration at which protease inhibitors displayed these effects should be mentioned.

We added this information (line 22)

  1. “Since recent experiments have indicated that the sensitivity of SARS-CoV-2 to antiviral compounds depends on the employed cell line”. Reference needed.

We added the reference (line 75)

III.        In Table 1 footnote: “Tox: substance showed toxicity” and in the following sentence: “Concentrations affecting cell growth or catabolism were excluded from further analyses (Table 1). The highest sensitivity was observed with Calu-3 and Huh-7 cells, where six more compounds could not be used at concentrations of 10 µM or more, indicating that Vero cells tolerate higher concentrations (Table 1)”. Admittedly, it is reliable to exclude the cytotoxic effects of the studied compounds during the evaluation of their viral replication inhibitory activity. However, the authors have to clarify the cytotoxicity threshold (growth inhibition% per µM concentration of the tested compound) above which they considered “substance showed toxicity”.

We added this information (line 91)

  1. Materials and Methods: “The cells were incubated with the compounds and subsequently infected with SARS-CoV-2. After 72 h ………” The time between incubating the compounds and infection with SARS-CoV-2 should be clarified. Was it the same as the time of incubation of the compounds for the preliminary cytotoxicity study??

We added this information (lines 267-268).

  1. The viral replication was determined by RT-qPCR (Figure 2). There are two figures 1, please renumber the second figure.

We corrected this mistake.

  1. The results (e.g. “The pyridin-3-yl indole-carboxylates 1 and 2 were toxic to Calu-3 cells but not to the other two cell lines. The protease inhibitors 4 and 5, which contain a fused thiophene or furan ring, failed to suppress SARS-CoV-2 …… Disulfiram, used as a control ……”) should be more connected, more determined and more thoroughly discussed. Referral to relevant figures or tables is preferred.

We added the referral. We added reference data and a new reference

VII.        Summary: “In Calu-3 cells, viral replication was less susceptible to inhibition by several compounds ……. If Calu-3 cells exhibit a lower drug uptake, which resembles the in vivo tissue ….”. Is there any evidence or reference to be cited which supports the hypothesis of “Calu-3 cells exhibit a lower drug uptake”?

This hypothesis is currently not proven. -> we stated this now in the manuscript

VIII.        “Vero and Huh-7 cells were seeded in optical 96-well plates. The following day, the cell numbers per individual well 75 were determined with an Insight reader ……….. (MTT) on these cell lines and on Calu-3 cells as described previously. [6-9]”. Detailed experimental procedures should be transferred to the Materials and Methods section.

We added this information.

  1. Re-editing for clarity enhancement may be beneficial, e.g. “The medium was replaced 24 h after infection by a medium containing the respective test compounds to remove defective viruses”.

We added and rewrote the description (lanes 115-116 & 250-251)

  1. As expected, the observed suppression of viral replication did not correlate with the IC50values determined in the in vitro enzyme assays (Table 1)”. Refer to the relevant Figure along with Table 1.

We added the reference to the Figure.

Reviewer 3 Report

The manuscript "Cell type-specific anti-viral effects of novel SARS-CoV-2 main protease inhibitors" addresses an important issue: the impact of novel pyridyl indole esters and peptidomimetics on viral replication. The authors conducted an in vitro study to determine whether those compounds have the potential to block SARS-CoV-2 in cell culture.

Unfortunately, the work presents many methodological errors that invalidate the conclusions. Therefore, I do not recommend the publication of this manuscript. Quantitative analysis of the experimental data is missing throughout the manuscript and the interpretations of the results and the discussion are thus suffering from these limitations.

The paper is interesting but there is a need for more experimental detail in order to critically review the data. The article propose some premises, however the results should be considered as a very preliminary that need much more experiments to provide a constructive conclusion.

Major points:

1.      Figures 1 and 1 (both are named equally) are incorrectly named, they are insufficient and lack of sufficient quality controls. Their caption is scarce and a more detailed description is needed.

2.      A more thorough description of the previously designed compounds and their mechanism of action is needed in the Introduction section.

3.      Unify the style of the references in the References Section and add DOI in the cases it is possible. And use the same reference and citation (follow MDPI’s guidelines) style in the main text. The references are insufficient and almost half of them are works from the authors of the manuscript.

4.      The Methods section in the study should be more accurately and thoroughly described for each technique used.

Author Response

  1. Figures 1 and 1 (both are named equally) are incorrectly named, they are insufficient and lack of sufficient quality controls. Their caption is scarce and a more detailed description is needed.

We corrected the numbering and added a new scheme to Figure 2

  1. A more thorough description of the previously designed compounds and their mechanism of action is needed in the Introduction section.

We added this information and new references.

  1. Unify the style of the references in the References Section and add DOI in the cases it is possible. And use the same reference and citation (follow MDPI’s guidelines) style in the main text. The references are insufficient and almost half of them are works from the authors of the manuscript.

We corrected the style and added new references.

  1. 4.The Methods section in the study should be more accurately and thoroughly described for each technique used.

We extended the Method section.

Round 2

Reviewer 3 Report

The authors have modified the former version of the manuscript according to the comments and suggestions presented.